# Nutritional Therapy Modulates Intestinal Microbiota and Reduces Serum Levels of Total and Free Indoxyl Sulfate and P-Cresyl Sulfate in Chronic Kidney Disease (Medika Study)

**DOI:** 10.3390/jcm8091424

**Published:** 2019-09-10

**Authors:** Biagio Raffaele Di Iorio, Maria Teresa Rocchetti, Maria De Angelis, Carmela Cosola, Stefania Marzocco, Lucia Di Micco, Ighli di Bari, Matteo Accetturo, Mirco Vacca, Marco Gobbetti, Mattia Di Iorio, Antonio Bellasi, Loreto Gesualdo

**Affiliations:** 1Nephrology, “A. Landolfi” Hospital, 83029 Solofra, Italy; br.diiorio@gmail.com (B.R.D.I.);; 2Department of Emergency and Organ Transplantation, Nephrology, Dialysis and Transplantation Unit, University of Bari Aldo Moro, Piazza G. Cesare 11, 70124 Bari, Italy; 3Department of Soil, Plant and Food Science, “Aldo Moro” University, Bari, Via G. Amendola 165/a, 70126 Bari, Italy; 4Department of Pharmacy, University of Salerno, 84084 Fisciano, Italy; 5Faculty of Science and Technology, Free University of Bozen, 39100 Bolzano, Italy; 6Data Scientist, Landolfi Nephrology Consultant, 83100 Avellino, Italy; 7Department of Research, Innovation and Brand Reputation, ASST Papa Giovanni XXIII, 24121 Bergamo, Italy

**Keywords:** CKD, microbiome, indoxyl sulfate, P-cresyl sulfate, very low protein diet, Mediterranean diet

## Abstract

In chronic kidney disease (CKD), the gut-microbiota metabolites indoxyl sulfate (IS) and p-cresyl sulfate (PCS) progressively accumulate due to their high albumin-binding capacity, leading to clinical complications. In a prospective crossover controlled trial, 60 patients with CKD grades 3B–4 (GFR = 21.6 ± 13.2 mL/min) were randomly assigned to two dietary regimens: (i) 3 months of free diet (FD) (FD is the diet usually used by the patient before being enrolled in the Medika study), 6 months of very low protein diet (VLPD), 3 months of FD and 6 months of Mediterranean diet (MD); (ii) 3 months of FD, 6 months of MD, 3 months of FD, and 6 months of VLPD. VLPD reduced inflammatory Proteobacteria and increased Actinobacteria phyla. MD and VLPD increased some butyrate-forming species of Lachnospiraceae, Ruminococcaceae, Prevotellaceae, Bifidobacteriaceae, and decrease the pathobionts Enterobacteriaceae. The increased level of potential anti-inflammatory *Blautia* and *Faecalibacterium*, as well as butyrate-forming *Coprococcus* and *Roseburia* species in VLPD was positively associated with dietary intakes and it was negatively correlated with IS and PCS. Compared to FD and MD, VLPD showed a lower amount of some *Lactobacillus*, *Akkermansia*, *Streptococcus*, and *Escherichia* species. MD and VLPD reduced both the total and free serum IS (MD −36%, −40% and VLPD −69%, −73%, respectively) and PCS (MD −38%, −44% and VLPD −58%, −71%, respectively) compared to FD. VLPD reduced serum D-lactate compared to MD and FD. MD and, to a greater extent, VLPD are effective in the beneficial modulation of gut microbiota, reducing IS and PCS serum levels, and restoring intestinal permeability in CKD patients.

## 1. Introduction

Over time, nutritional therapy (NT) has become a cornerstone of the conservative treatment of chronic kidney disease (CKD), aiming basically to administrate a reduced amount of high biological value proteins to reduce the urea production in patients with CKD [1,2]. There are studies showing that NT induces favorable metabolic changes, delaying the progression of the disease and the need for dialysis by preventing signs and symptoms of renal failure [1,2,3].

Recently, the focus has moved from a low-protein nutritional therapy, which favors the protein quantity of the diet, to a qualitative evaluation of the proteins prescribed to the patients with CKD [4] in order to increase the dietary fiber content [5,6,7,8], with the aim of enabling a correct and physiological management of the intestinal microbiome [9] and in order to reduce urea levels [10]. The Mediterranean diet (MD), which covers the daily needs of nutrients and proteins recommended for the general population, is useful in the early stages of CKD [5,11,12]. However, as CKD progresses, a lower protein intake becomes increasingly necessary, up to regimens that include a vegan diet [13,14] or a very low protein diet (VLPD) supplemented with essential amino acids and ketoacids [15,16]. The latter are capable of inducing a massive reduction of urea to levels that may be deemed comparable to healthy subjects [10]. Urea and the uremic milieu contribute indeed to alterations of gut microbiota and gut integrity, and to increased uremic toxins production, in a vicious circle [17]. Microbiota and its metabolites, particularly protein-bound uremic toxins such as indoxyl sulfate (IS) and p-cresyl sulfate (PCS), seem to play an increasing role in the incidence of cardiovascular disease in CKD, as well as in blood pressure regulation and hypertension [18]. Due to the high binding affinity of IS and PCS to albumin, they cannot be efficiently removed by conventional hemodialysis and progressively accumulate in CKD patients leading to disease progression and resulting in organ damage [18]. Therefore, strategies aimed at lowering their intestinal production are desirable.

## 2. Experimental Section

### 2.1. Study Design

This prospective, randomized, crossover control trial involving 60 CKD grade 3B–4 incident patients has been rigorously described in our previous work [10]. The local Ethics Committee (Campania Nord Ethics Committee, Avellino, Italy) approved this study after ascertaining its compliance with the dictates of the Declaration of Helsinki (IV Adaptation). All participants signed their informed consent. The study protocol was registered on ClinicalTrials.gov with the identifier number NCT02302287. Study design (Appendix A), dietary regimens, timing of sample collection and medical examination, NT adherence have been reported in our previous work [10] and have been described in Appendix A. The changes in fecal microbiota by MD and VLPD are the primary endpoint; the change in microbial-derived IS and PCS levels due to the effects of MD and VLPD is a secondary endpoint, together with the change in renal function as well as nutritional and inflammatory status (i.e., urea, eGFR, BUN, acid–base balance, serum creatinine, serum and urinary electrolytes, blood pressure, parathyroid hormone). Software randomization divided CKD patients into two treatment arms (A and B), with a 1:1 ratio (Figure 1). Group A: 30 patients who alternately followed 3 diet regimens as described—3 months of FD (FD is the diet usually used by the patient before being enrolled in the Medika study), 6 months of VLPD, 3 months of FD, and 6 months of MD. Group B: 30 patients who alternately followed 3 diet regimens as described—3 months of FD, 6 months of MD, 3 months of FD, and 6 months of VLPD. The three different dietary regimes (Table 1) have previously been fully described [10]. In short, VLPD involves the administration of one tablet every 5 kg of patient’s ideal body weight, supplied during breakfast, lunch, and dinner. Each VLPD tablet (alphaKappa, Fresenius Kabi) contains L-lysine, L-threonine, L-histidine, L-tyrosine, keto-isoleucine, keto-leucine, keto-phenylalanine, keto-valine, hydroxy-methionine [19]. Thus, for every kg of body weight, 285 mg of essential amino acids and 323 mg of ketoanalogs were orally administered to the patients. The ketoanalogs were administered in a range of 8–18 cpr corresponding to a mean ± SD of 14.2 ± 2.5 cpr/patients.

Each crossover timing involved 3 months of FD as a washout period to avoid any potential carry-over effect of MD or VLPD. NT adherence was evaluated by dietary interviews completed by patients for a full week every two months and by photos of the dishes they would eat. The protein intake was estimated by means of the Maroni formula [20] based on 24 h urine collections. The adherence to MD was evaluated by the Mediterranean diet score [21], which is a score composed of 9 items, each worth a point. All the study participants had a score >4 (medium), and 6 of them had a score of 9 (maximum). Blood, stools, and urinary samples from each patient were collected on enrollment (T0), and after 3 (T3), 9 (T9), 12 (T12), 18 (T18) months after the beginning of the study. With the same timing, the patients also underwent a medical examination. The expected outcomes were a new balance in intestinal flora, favoring a beneficial shift to a saccharolytic metabolism and a decrease in PCS and IS production as well as of inflammatory status. We also evaluated risk analysis and possible problems and solutions that have been reported in Appendix A.

### 2.2. Collection of Serum and Fecal Samples

Serum and fecal samples were collected at the end of each diet regimen with an average of two groups. Serum was collected from total blood samples, after centrifugation at 2300 rpm at 4 °C for 15 min and was stored at −80 °C until use. After collection, fecal samples were immediately mixed with RNAlater (Sigma-Aldrich, St. Louis, MO, USA) (ca. 5 g, 1:2 (wt/vol)) or Amies transport medium (Oxoid Ltd., Basingstoke, Hampshire, UK) (ca. 15 g, 1:1 (wt/wt)) under anaerobic conditions (AnaeroGen, Oxoid Ltd., Basingstoke, Hampshire, UK), and stored at −80 °C.

### 2.3. Reagents

Indoxyl sulfate potassium salt, ammonium acetate of mass spectrometry grade, methanol, acetonitrile, and distilled water of high-performance liquid chromatography (Ultra CHROMASOLV) grade were purchased from Sigma (St. Louis, MO, USA), p-cresyl sulfate ammonium salt was from ALSACHIM (Bioparc, Illkirch, France) and indoxyl-4,5,6,7-d4 sulfate potassium salt was from Toronto Research Chemicals (North York, ON, Canada).

### 2.4. LC-MS/MS for Quantification of pCS and IS

Circulating levels of IS and pCS were determined at the end of a washout period of 3 months of FD (baseline) and at the end of each dietary regimens, for a total of 3 dosages for each patient. Total and free IS and pCS serum levels were assayed by mass spectrometry analysis as previously described [22]. All samples were processed in duplicate and run at least in duplicate to reach an intra-assay coefficient of variation <10%.

### 2.5. Serum D-Lactate Measurement

The analysis of the D-lactate was performed using a colorimetric enzymatic assay (AAT Bioquest, Sunnyvale, CA, USA) following the manufacturer’s instructions. Briefly, a calibration curve was prepared from serial dilutions from 1 mM D-lactate standard solution to get 300, 100, 30, 10, 3, and 1 μM plus a blank control. Serial dilutions of D-lactate standard (50 μL) and test samples (50 μL), were added in duplicate into a white clear bottom 96-well microplate. Later, a D-lactate assay mixture (50 μL), composed of enzyme mix, assay buffer, and NAD stock solution 100×, was added to each well of D-lactate standard, blank control, and test samples to obtain a total D-lactate assay volume of 100 µL/well. The reaction mix was incubated at room temperature, protected from light, and the absorbance was monitored after 90 min with an absorbance plate reader at A540 nm.

### 2.6. RNA Extraction from Fecal Samples and 16S Metagenomic Sequencing

An aliquot of 200 mg of fecal sample diluted in RNAlater was used for RNA extraction with the Stool Total RNA Purification Kit (Norgen Biotek Corp., Thorold, ON, Canada). The resulting total RNA (ca. 2.5 μg) was transcribed to cDNA using random hexamers and the Tetro cDNA synthesis kit (Bioline USA, Inc., Taunton, MA, USA), according to the manufacturer’s instructions. Libraries for NGS were prepared starting from 12.5 ng of microbial genomic cDNA, as described in the Illumina 16S Metagenomic Sequencing Library Preparation guide. Primers for target enrichment were selected from Klindworth [23] in order to amplify and, conversely, sequence the V1–V3 region of the 16S rRNA. The enrichment of the target region was achieved through the following first PCR amplification step: preheating at 95 °C for 3 min; 25 cycles of denaturation at 95 °C for 30 s, annealing at 55 °C for 30 s, an extension at 72 °C for 30 s; terminal extension at 72 °C for 5 min. The amplified DNA was visualized using Bioanalyzer DNA 1000 chip (Agilent Technologies Inc., Santa Clara, CA, USA) to verify the size (about 500–600 bp) and the quality of the libraries. A second PCR amplification step was performed to attach Illumina dual indices and sequencing adapters. Finally, sequencing was performed on the Illumina MiSeq desktop sequencer, using paired 300 bp reads, and MiSeq v3 reagents (Illumina, San Diego, CA, USA). The sequences were first clustered into operational taxonomic unit clusters with 97% identity (3% divergence), using USEARCH. To determine the identities of bacteria, sequences were first queried, using a distributed BLASTn.NET algorithm24 against 16S bacterial sequences, which were derived from NCBI. Database sequences were characterized as high quality on the basis of criteria that were originally described by the Ribosomal Database Project (RDP, version 10.28).

### 2.7. Statistic

Since no data were present in the current literature regarding the effects of diet regimens, no sample size estimation was carried out. Variables were presented as the mean ± standard deviation (SD), median (interquartile range), or count (percentage) as appropriate. Differences between diet regimens were evaluated by Student’s *t*-test or Wilcoxon rank sum test for normally and not-normally distributed variables, respectively, as appropriate. All analyses were conducted as intention-to-treat. Two-tailed *p* < 0.05 values were considered statistically significant. Analyses were completed using R version 3.1.3 (2015-03-09) (The R Foundation for Statistical Computing, Vienna, Austria). StatView software package SAS, 5.0 version, was used for linear regression analysis determining the correlation between variables. Metagenomic data (UniFrac distance metric and taxonomic abundance) were analyzed by principal component analysis (PCA) [24,25] using the statistical software Statistica for Windows (Statistica 6.0 for Windows 1998, StatSoft, Vigonza, Italy). Measures of diversity were screened for group differences by using analysis of variance (ANOVA). PermutMatrixEN software was used to identify clusters at the level of the sample groups and taxa [26]. Spearman correlations between OTU and metabolite concentration were computed. All analyses were conducted in R, using the vegan, labdsv, DESeq2, and phyloseq packages.

## 3. Results

### 3.1. Patients

All 60 CKD patients completed the study. The anthropometric, clinical, and biochemical data of the enrolled patients according to each nutritional intervention have been reported in Table 1. The extra treatments, as well as the interruption and resumption of treatments of our cohort of patients have been comprehensively described in our previous work [10].

In brief, VLPD was more effective than MD in lowering the systolic and diastolic blood pressure and the serum levels of urea, sodium, phosphorus, and parathyroid hormone. VLPD was associated with increased serum bicarbonate and hemoglobin levels (Table 2) [10]. Finally, MD and VLPD lowered urinary sodium and potassium levels, but only VLPD lowered urinary urea nitrogen levels, the fractional excretion of phosphorus and sodium intake according to literature data [2,10,27,28].

### 3.2. Microbial Components Linked to Specific Dietary Intake

The three different dietary regimens did not affect the microbial alpha diversity values (data not shown). Similarly, there was no clear separation by microbiome composition in FD, MD, and VLPD samples in PCA plots based on UniFrac distances. However, at phylum level, Firmicutes were detected at lower level during FD compared to the other diets (*p* < 0.004) (Figure 2). The level of Bacteroidetes was lower in MD than FD (5.9% vs. 9.6%, *p* = 0.026). The VLPD strongly reduced the relative amount of Proteobacteria compared to the other diet (*p* < 0.012). Compared to MD, VLPD harbored higher level of Actinobacteria (*p* = 0.028). At the same time, VLPD showed the lowest relative abundance (*p* < 0.030) of Verrucomicrobia. VLPD harbored the highest levels of Lachnospiraceae (34.4%, *p* < 0.006), Ruminococcaceae (19.6%, *p* < 0.013), Prevotellaceae (3.1%, *p* < 0.029), and Bifidobacteriaceae (4.2%, *p* < 0.009) (Figure 2B). Compared to FD, MD showed a strong increase of Lactobacillaceae (12.7% vs. 5.2%, respectively, *p* = 0.047) together with a decrease of the relative amount of Bacteroidaceae (2.3% vs. 0.67%, respectively, *p* = 0.026) and Coriobacteriaceae (1.9% vs. 4.1%, respectively, *p* = 0.009). Compared to FD and MD diets, VLPD showed lower amount of Lactobacillaceae (*p* < 0.047); Streptococcaceae (*p* < 0.015), Verrucomicrobiaceae (*p* < 0.028), and Enterobacteriaceae (*p* < 0.030). At genus and species levels, VLPD harbored the highest relative amount of *Blautia* (*B. coccoides*, *B. hydrogenotrophica*, *B. obeum*, and *B. wexlerae*), *Bifidobacterium* (*B. adolescentis* and *B. pseudolongum*), *Clostridium (C. cadaveris)*, *Faecalibacterium* (*F. prausnitzii*), *Coprococcus* (*C. eutactus*), and *Roseburia* (*R. faecis*) (Figure 3 and Table 3). Compared to FD, both MD and VLPD resulted in a decrease of *Ruminococcus* (*R. callidus*), *Collinsella* (*C. aerofaciens*), and *Bacteroides* (*B. uniformis* and *B. vulgatus*). Compared to FD and/or MD, VLPD showed a lower relative amount of the genera *Lactobacillus* (e.g., *L. gasseri*, *L. salivarius*), *Akkermansia* (*A. muciniphila*), *Streptococcus* (*S. bovis*, *S. mutans*, *S. sobrinus*, and *S. vestibularis*) and *Escherichia* (*E. albertii*). The genus *Enterococcus* (*E. lactis*) was mainly associated with the MD diet.

### 3.3. Correlations between Diet and Microbiome

There was a very strong relationship between fecal microbiome profile and nutrient intake (Figure 4). Several species (e.g., *Akkermansia muciniphila*, *Clostridium perfringens*, *Bacteroides eggerthii*, *Escherichia albertii*, *Parabacteroides merdae*, *Streptococcus vestibularis*, *Streptococcus parasanguinis*, *Serratia entomophila*, *Eubacterium cylindroides*, *Ruminococcus callidus*, *Anaerobranca zavarzinii*, and *Bacteroides vulgatus*) were positively correlated (*r >* 0.7; FDR < 0.05) with total and animal proteins, lipids, sodium, calcium, and phosphorus intakes. The same species showed negative correlations with the intake of dietary fibers, vegetal proteins, potassium, and ketoanalogs. Opposite correlation trends were found for other dominant species (e.g., *Faecalibacterium prausnitzii*, *Roseburia faecis*, *Blautia wexlerae*, *Blautia hydrogenotrophica*, *Blautia obeum*, *Bifidobacterium pseudolongum*, *Sarcina maxima*, *Lachnospira pectinoachiza*, *Coprococcus eutactus*, *Blautia coccoides*, *Pseudobutyrivibrio xylanivorans*, and *Clostridium cadaveris*).

### 3.4. Uremic Toxins

VLPD is very effective in reducing both total and free IS (2.8 ± 2.8 µg/mL, *p* < 0.0001; 0.08 ± 0.12 µg/mL, *p* < 0.0001) and PCS (10.1 ± 8.8 µg/mL, *p* < 0.0001; 0.28 ± 0.4 µg/mL, *p* < 0.0001) serum levels as compared to FD (IS: 9.2 ± 6.7 µg/mL; 0.3 ± 0.3 µg/mL. PCS: 24.0 ± 13.4 µg/mL; 0.97 ± 0.98 µg/mL). MD is also able to decrease the serum levels of total and free IS (5.9 ± 6.1 µg/mL, *p* = 0.002; 0.18 ± 0.2 µg/mL, *p* = 0.003) and PCS (14.8 ± 9.4 µg/mL, *p* < 0.0001; 0.54 ± 0.6 µg/mL, *p* = 0.001) as compared to FD, although to a lesser extent than s-VLPD (Figure 5). Total and free IS and PCS serum levels were positively correlated with azotemia (*r* = 0.44, *p* < 0.0001; *r* = 0.46, *p* < 0.0001; *r* = 0.37, *p* < 0.0001; *r* = 0.39, *p* < 0.0001, total and free, respectively) (Figure 6). A wider variation of the azotemia, total IS and PCS between VLPD and FD compared to that between MD and FD was found, as well as a mild positive correlation between the variation of azotemia, and the variation of total IS (*r* = 0.2, *p* < 0.03) and PCS (*r* = 0.22, *p* < 0.01).

Furthermore, both total and free IS and PCS serum levels were positively correlated with protein intake (*r* = 0.42; *r* = 0.48; *r* = 0.48; *r* = 0.42, *p* < 0.0001 for all, total and free respectively) and PTH (*r* = 0.34, *p* < 0.0002; *r* = 0.30, *p* < 0.0001; *r* = 0.14, *p* < 0.003; *r* = 0.25, *p* < 0.002, total and free respectively), while were negatively correlated with the creatinine clearance values (*r* = −0.27, *p* = 0.0005; *r* = −0.28, *p* = 0.0005; *r* = −0.24, *p* = 0.002; *r* = −0.34, *p* < 0.0001, total and free respectively) of the enrolled patients.

### 3.5. VLPD Reduces Intestinal Permeability, with Decreasing Blood Urea Nitrogen

Circulating D-lactate was measured in serum samples of patients following FD, MD, and VLPD, as a marker of intestinal permeability. VLPD significantly reduced D-lactate in CKD patients (188.3 μM (83.0–364.1 μM)), in comparison with FD [542.3 μM (368.1–836.1 μM), *p* = 0.001] and MD (448.5 μM (225.3–931.3 μM), *p* = 0.04)). MD treatment showed significant variation of D-lactate levels in comparison with FD. D-Lactate levels directly correlated (*r* = 0.3, *p* = 0.002) with blood urea nitrogen levels (Figure 7).

### 3.6. Correlations between Microbiome and Metabolome

*E. albertii* was positively correlated (*r >* 0.40; FDR < 0.05) with free and total IS and PCS serum levels. On the contrary, *Blautia coccoides* was negatively associated with (*r* < −0.20; FDR < 0.05) free and total IS and PCS serum levels. *Roseburia faecis*, *Clostridium cadaveris*, and *Faecalibacterium prausnitzii* were also negatively associated with the total IS and PCS (Figure 8).

## 4. Discussion

To date, Medika is the first known study evaluating the effects of MD and VLPD on the modulation of intestinal microbiota and, consequently, on the variation of serum IS and PCS levels in patients with CKD. In CKD, a variety of uremic toxins, including IS and PCS, accumulate in the blood leading to clinical complications [18]. IS and PCS are produced at the intestinal level by proteolytic microbes and are finally excreted in the urine by active tubular secretion. Their clearance is affected by kidney excretory capacity, indeed, in CKD, as kidney function declines, they progressively accumulate in blood and they cannot be removed by conventional dialysis due to their strong albumin binding [18].

Based on this evidence, NT represent an effective strategy to manipulate the intestinal microbiome in CKD patients. NT has already been demonstrated to have favorable physiological implications in the treatment of CKD patients, reducing the use of drugs by promoting pharmacological compliance [2,8,19,21,28].

Our results showed that at baseline, under FD, a lower relative abundance of Bacteroidetes and strong gut colonization of Firmicutes phyla in CKD patients when compared to the healthy population [24,29]. According to the literature, dysbiosis in CKD patients includes an increased abundance of Proteobacteria [30], which is associated to inflammation and/or facilitates gut colonization by exogenous pathogens [31], and the lowest level of Actinobacteria [32]. Firstly, this study shows that VLPD was effective in reducing the Proteobacteria and improving the gut colonization of Actinobacteria in CKD patients. Ruminococcaceae and Lachnospiraceae, producing butyric acid, seem to be protective against inflammatory bowel disease and, indeed, they were positively correlated to healthy state [33]. The gut microbiome of CKD patients resulted in a strong decrease of butyrate-forming species such as *Roseburia* spp. and *Faecalibacterium prausnitzii* [17,34]. The positive effect of butyric acid on colonic inflammation is well known [30]. MD and especially VLPD resulted in an increase of Lachnospiraceae, Ruminococcaceae, Prevotellaceae, and Bifidobacteriaceae together with a strong decrease of Enterobacteriaceae in fecal samples of CKD. At colon level, some species of Enterobacteriaceae are considered pathobionts [35] and, consequently, the overgrowth of Enterobacteriaceae in the gut of CKD patients may further promote a metabolic shift in intestinal lumen and potentiate the emergence of pathobionts [29]. Indole-forming and p-cresol-forming genera and species were also positively associated with ESRD patients [30]. Specific associations between species and dietary intake were found. Compared to MD, VLPD was more effective in improving the composition of bacterial genera and species in CKD patients. The increased level of *Blautia*, *Faecalibacterium*, *Coprococcus*, and *Roseburia* species in VLPD was positively associated with prescribed dietary fibers, vegetal proteins, potassium, and ketoanalogs, and it was negatively correlated with serum toxins in CKD patients. A positive anti-inflammatory effect of *Blautia* was described in other clinical settings [36]. *F. prausnitzii* showed anti-inflammatory effects in both animal and human subjects [37]. *Roseburia* and *Coprococcus* species were reported to be able to hydrolyze starch and other sugars producing butyric acid and other SCFA [38]. High fecal *Bacteroides*, *Ruminococcus*, *and Streptococcus* abundance was positively associated with protein and animal fat diet [39]. Accordingly, this study showed that some *Bacteroides*, *Ruminococcus*, *Streptococcus*, *Akkermansia*, and *Escherichia* species (e.g., *Bacteroides vulgatus*) were negatively associated to VLPD. It was shown that *B. vulgatus* is able to colonize and persist in inflammatory bowel diseases (IBD) patients [40]. Some genera such as *Escherichia* and *Proteus* were previously found in the blood of ESRD patients. Accordingly, this study shows positive association between *E. albertii* and the free and total IS and PCS serum levels. Previously, *A. muciniphila* was positively correlated with healthy status protecting against numerous diseases [41]. A study showed that NLRP6, an innate immune receptor, is important for suppressing the development of spontaneous colitis in the IL10^−/−^ mouse model of IBD [42]. Moreover, NLRP6-deficiency causes an enrichment of *A. muciniphila* population, which can act as a pathobiont to promote colitis in a genetically susceptible host. The controversial role of *A. muciniphila* in the development of inflammatory intestinal disease will be probably worked up in future studies, possibly leading to promising therapies targeted to mucosa-associated bacterial composition. In this study, *A. muciniphila* was significantly less abundant in VLPD indicating the highest colonizing activity in patients treated with protein- and fat-rich diets.

Along with modulating microbiota composition, MD, and VLPD, reducing the protein intake allows a significant reduction of urea with concomitant reduction of IS and PCS levels. This was further supported by the expected positive association found between protein intake and IS and PCS serum levels as well as for azotemia and their corresponding variations. Consistently with our data, a previous study showed that the use of VLPD induced a reduction of IS compared to low protein diet in CKD patients [43]. While, a recent study showed a decreased in PCS but not IS after 6 months of low protein diet [44]. On the contrary, to date, no studies have been conducted on CKD patients that demonstrate the efficacy of MD on the reduction of IS and PCS serum levels. Only one recent study showed no association between plasma levels of IS and PCS with a Mediterranean diet score or with intake of dietary precursors [45]. The negative association between IS and PCS serum levels and creatinine clearance found in our patients could help to clarify the relationship between increased uremic toxins levels and CKD progression that has not yet been conclusively established [18]. On the contrary, the positive correlation between PTH and serum IS and PCS was unexpected because it has been shown that higher levels of uremic toxins reduced in vitro bone formation and bone resorption [46] as opposed to the effects of PTH. Recent research suggested that PCS and IS could also be involved in bone and mineral disorders in patients with CKD, modulating the effects of PTH in bone turnover. However, such relationships are very complex and have not been fully elucidated [47]. Since the increased IS and PCS serum levels in CKD causes the PTH resistance [48], the bone and renal responses to the action of the PTH are progressively decreased. Indeed, it has been hypothesized that the effects of IS and PCS levels in CKD-BMD depends on the CKD stage and on PTH levels [49]: the increasing IS and PCS serum levels at early stage of CKD decrease the PTH signaling expression contributing to low bone turnover disorders; in late CKD stage, IS and PCS accumulate along with persistent PTH secretion (which does not fully express its activity) contributing, alongside other factors, to PTH effect on high bone turnover disorders [49]. Therefore, VLPD, decreasing IS, PCS, and PTH may also contribute to improve bone health in CKD patients.

Our data showed the reduction of plasma D-lactate following VLPD indicating a reduction of intestinal permeability. This could probably be due to the ability of VLPD to control uremia and to decrease circulating and colonic urea levels. The positive correlation found between circulating D-lactate levels and azotemia seems to further confirm this hypothesis. Intestinal permeability is known to contribute to systemic inflammation. In addition to the increased production of uremic toxins, their translocation together with bacterial LPS through the epithelial intestinal barrier contributes to the local and systemic inflammation that accompanies the CKD [50,51,52,53,54,55]. In this context, VLPD seems to represent an anti-inflammatory diet as the observed decrease in CRP in this study demonstrates. However, the aim of the present work was not to investigate the direct anti-inflammatory effect of NT which will certainly require future investigation. The strengths of this study are the crossover methodology, the considerable number of patients who followed the alternation of different nutritional regimens, and the simultaneous observation of the fecal microbiota and its serum metabolites. In conclusion, the trial with MD and VLPD has been successful in modulating intestinal microbiota and in reducing the protein-bound uremic toxin production rate in CKD patients.

In conclusion, the Medika study irrefutably demonstrates that it is possible to manipulate the intestinal microbiome with the TN capable of reducing the values of azotemia below 100 mg/dL [56].

## Figures and Tables

**Figure 1 jcm-08-01424-f001:**
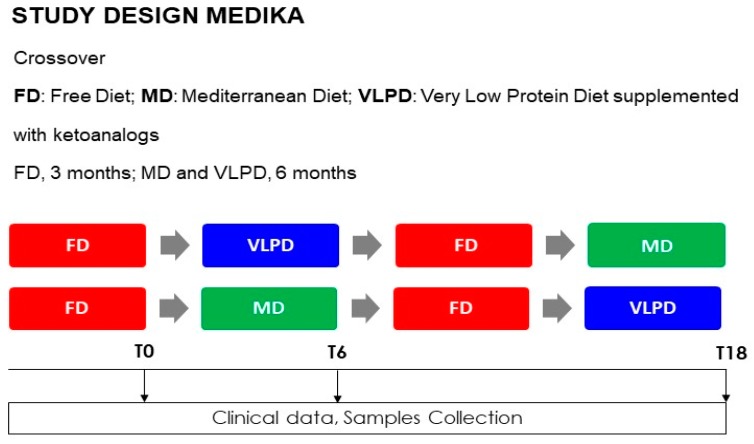
Study design: FD = free diet, VLPD = very low protein diet, MD = Mediterranean diet.

**Figure 2 jcm-08-01424-f002:**
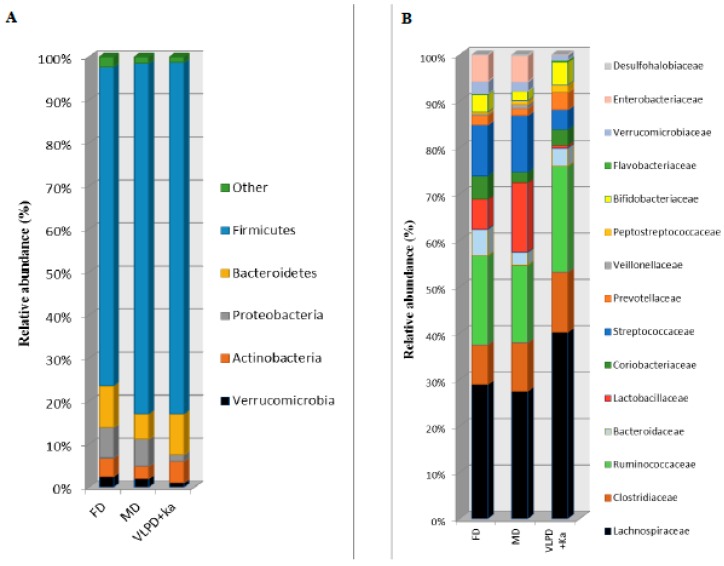
Bacterial phyla and families associated to dietary regimens in intestinal microbiota of chronic kidney disease (CKD) patients. Average of relative abundance (%) of bacterial phyla (**A**) and families (**B**) which were found in the fecal samples of patients with free diet (FD), Mediterranean diet (MD), or very low protein diet supplemented with alphaKappa (VLPD + Ka). Only phyla showing statistical differences (*p* < 0.05) in at least one dietary regimen were reported.

**Figure 3 jcm-08-01424-f003:**
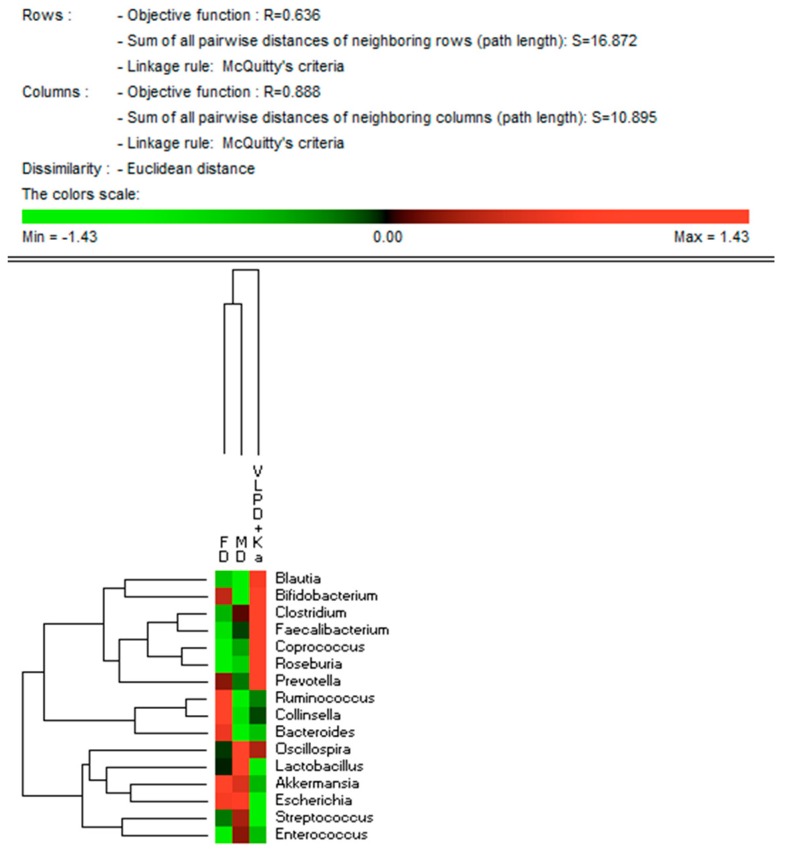
Bacterial genera associated to dietary regimens in intestinal microbiota of CKD patients. Average of relative abundance (%) of bacterial genera which were found in the fecal samples of patients with free diet (FD), Mediterranean diet (MD), or very low protein diet supplemented with alphaKappa (VLPD + Ka). Only genera showing statistical differences (*p* < 0.05) in at least one dietary regimen were reported.

**Figure 4 jcm-08-01424-f004:**
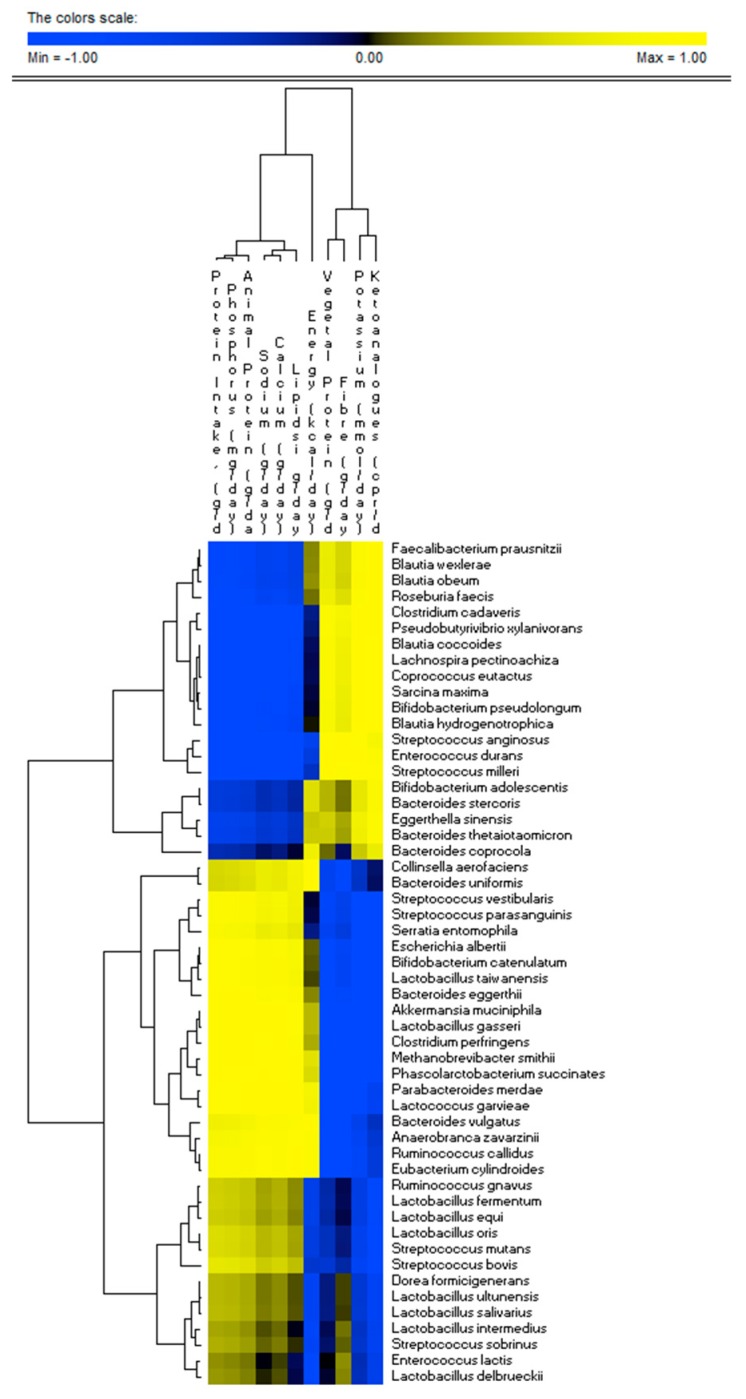
Correlations between bacterial species and dietary intake of CKD patients. The colors of the scale bar denote the nature of the correlation, with 1 indicating a perfectly positive correlation (yellow) and −1 indicating a perfectly negative correlation (blue) between species and dietary intake. Only significant correlations (FDR, 0.05) are shown.

**Figure 5 jcm-08-01424-f005:**
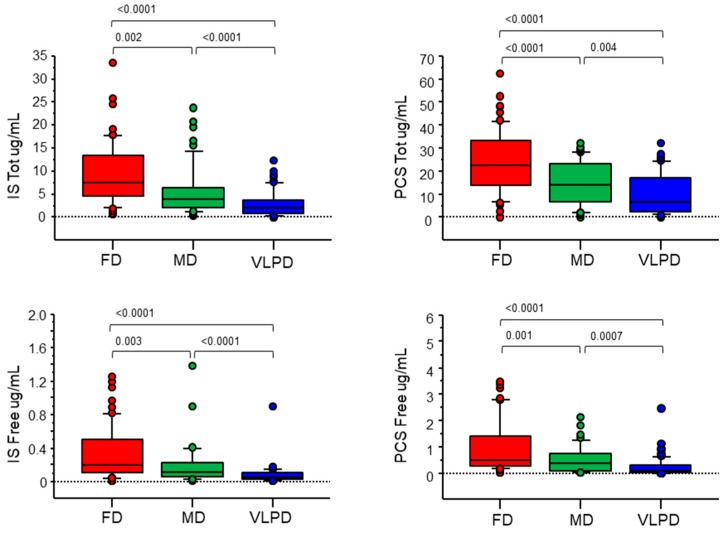
Serum levels of IS and PCS measured by MRM MS analysis. Serum levels of IS and PCS in CKD patients according to the different dietary regimens (Table 1). In the diagrams, the 25th and 75th percentiles are plotted as box, the 90th and 10th percentiles as bars, and observations less than the 10th percentile and greater than the 90th percentile as points. The central horizontal line within each box represents the 50th percentile. *p* values were calculated by Wilcoxon test.

**Figure 6 jcm-08-01424-f006:**
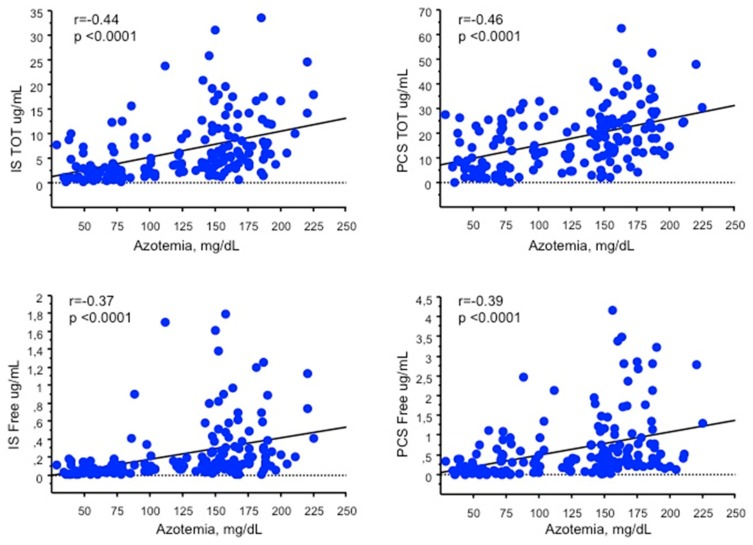
Correlations between the serum levels of (**A**) total indoxyl sulfate (IS TOT), (**B**) total p-cresyl sulfate (PCS TOT), (**C**) free indoxyl sulfate (IS free), (**D**) free p-cresyl sulfate (PCS free) and azotemia.

**Figure 7 jcm-08-01424-f007:**
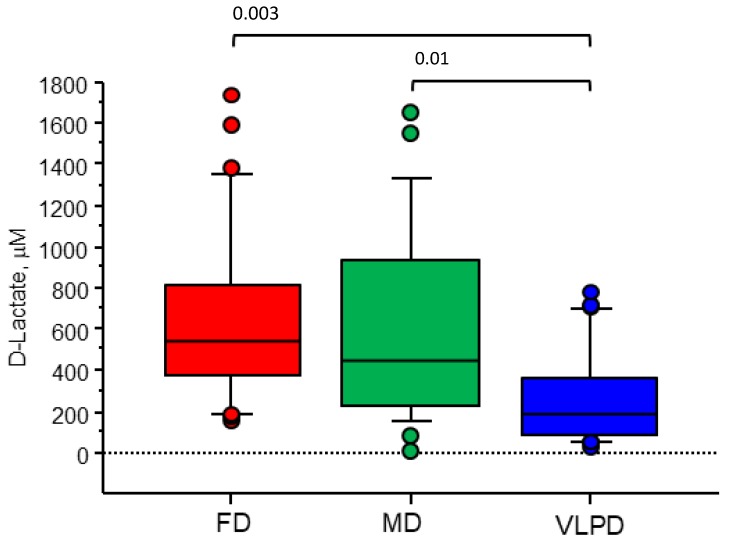
Serum levels of D-lactate measured in CKD patients according to dietary regimens (Wilcoxon test).

**Figure 8 jcm-08-01424-f008:**
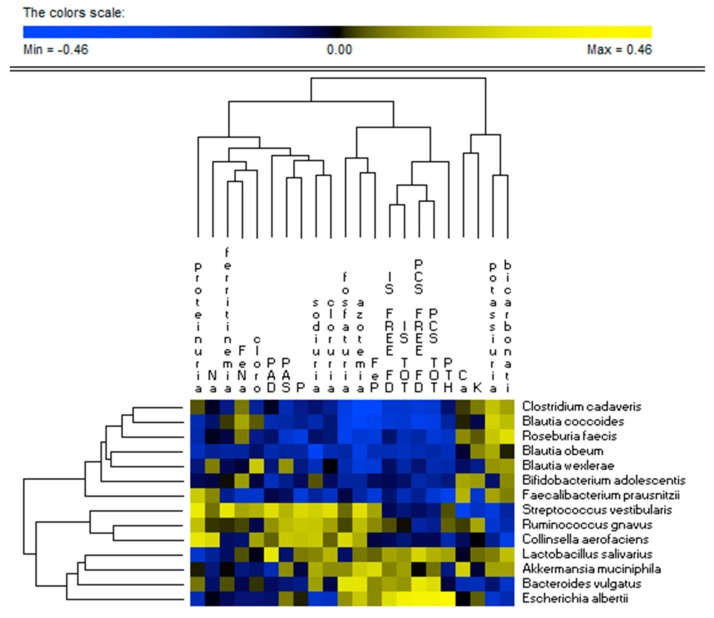
Correlations between bacterial species and serum metabolome of CKD patients. The colors of the scale bar denote the nature of the correlation, with 1 indicating a perfectly positive correlation (yellow) and −1 indicating a perfectly negative correlation (blue) between species and dietary intake. Only significant correlations (FDR, 0.05) are shown.

**Table 1 jcm-08-01424-t001:** Diet compositions.

	Mediterranean Diet	Very Low Protein Diet
Protein Intake (g/kg bw/day)	0.8	0.3
Animal Protein (g/day)	30–40	0
Vegetal Protein (g/day)	40–50	30–40
Energy (kcal/kg bw/day)	30–35	30–35
Sodium (g/day)	5–6	5–6
Potassium (g/day)	2–4	3–5
Calcium (g/day)	1.1–1.3	1.1–1.3
Phosphorus (g/day)	1.2–1.5	0.6–0.8
Ketoanalogs (cpr/5 kg bw/day)	0	1

**Table 2 jcm-08-01424-t002:** Anthropometric, clinical, and biochemical data of patients according to each dietary regimen.

Number	60			
Sex (M)	49			
Age	68.4 ± 12.3			
	Free Diet	Mediterranean Diet	VLPD	*p*
Body Mass Index	26.6 ± 3.9	26.9 ± 4.1	26.8 ± 4.0	0.887
SBP, mmHg	139 ± 20	135 ± 17	126 ± 12 *	0.001
DBP, mmHg	75 ± 12	76 ± 8	72 ± 6 *	0.047
Creatinine, mg/dL	3.83 ± 1.67	3.71 ± 1.39	3.48 ± 1.34	0.364
Urea, mg/dL	175.2 ± 22.7	136.7 ± 26.3 ^#^	68.3 ± 18.2 *	0.001
Glycemia, mg/dL	113 ± 31	112 ± 28	108 ± 22	0.579
Glycated hb (%) (*n* = 24 pts)	7.1 ± 1.1	7.3 ± 1.3	6.8 ± 1.1	0.079
Uricemia, mg/dL	5.9 ± 2.1	5.5 ± 1.6	5.1 ± 1.3	0.492
Natrium, mmol/L	140 ± 2	141 ± 2	138 ± 3 *	0.001
Potassium, mmol/L	5.0 ± 0.6	5.0 ± 0.6	4.9 ± 0.6	0.487
Calcium, mmol/L	9.2 ± 0.7	9.3 ± 0.9	9.2 ± 0.7	0.776
Phosphate, mg/dL	4.7 ± 0.9	4.3 ± 0.8 ^#^	3.6 ± 0.6 *	0.001
Bicarbonates, mmol/L	20.5 ± 2.96	22.9 ± 3.3 ^#^	25.1 ± 2.2 *	0.001
Cholesterol, mg/dL	151 ± 48	155 ± 38	156 ± 40	0.167
Triglycerides, mg/dL	135 ± 70	143 ± 63	141 ± 60	0.345
TSAT (%)	26 ± 6	27 ± 7	26 ± 8	0.517
Ferritin, ng/mL	144 ± 105	127 ± 132	120 ± 78	0.353
PTH, pg/mL	234 ± 162	232 ± 180	174 ± 97 *	0.001
Hemoglobin, g/dL	11.4 ± 1.6	11.6 ± 1.4	12.1 ± 0.7 *	0.001
Albumin, g/dL	3.6 ± 0.5	3.7 ± 0.4	3.7 ± 0.5	0.787
CRP, mg/L	6.2 ± 10.5	4.1 ± 4.4	2.6 ± 2.4	0.014
Diuresis, mL/day	2095 ± 415	2100±538	2095±380	0.153
Urinary natrium, mmol/day	164 ± 47	145 ± 41 *	123 ± 37 *	0.001
Urinary potassium, mmol/day	42 ± 14	46 ± 14	55 ± 15 *	0.001
Urinary phosphate, mmol/day	680 ± 177	507 ± 181 ^#^	298 ± 116 *	0.001
UUN, mmol/day	21 ± 6	17 ± 5 ^#^	9 ± 3 *	0.001
Prot-u, mg/day	1199 ± 1267	1212 ± 1094	963 ± 919*	0.001
Creatinine clearance, mL/min	21.6 ± 13.2	21.3 ± 12.0	21.8 ± 12.1	0.329
P intake, mg/day	971 ± 242	772 ± 231 ^#^	453 ± 171 *	0.001
Na intake, g/day	9.8 ± 2.1	8.5 ± 2.2 ^#^	7.4 ± 2.4 *	0.001

SBP: systolic blood pressure; DBP: diastolic blood pressure; PTH: parathyroid hormone; UUN: urinary urea nitrogen; CRP: C-reactive protein; TSAT: Transferrin Saturation. * Bonferroni test: *p* < 0.05 versus FD and MD. ^#^ Bonferroni test: *p* < 0.05 versus FD.

**Table 3 jcm-08-01424-t003:** Diet associated changes of microbiome in chronic kidney disease (CKD) patients. Average of relative abundance (%) of bacterial species differently found (*p* < 0.05) in feces of CKD patients after free diet (FD), Mediterranean diet (MD), and very low protein diet plus essential amino acids and ketoanalogs (ALFA-KAPPA) (VLPD + Ka).

Phylum/Family	Specie	FD	MD	VLPD + Ka	*p* FD vs. MD	*p* FD vs. VLPD + ka	*p* MD vs. VLPD + ka
Actinobacteria/Bifidobacteriaceae	*Bifidobacterium adolescentis*	1.22	0.56	1.72	0.037	0.227	0.020
	*Bifidobacterium catenulatum*	0.16	0.15	0.00	0.458	0.113	0.007
	*Bifidobacterium pseudolongum*	0.00	0.00	0.28	-	0.034	0.034
Actinobacteria/Coriobacteriaceae	*Collinsella aerofaciens*	2.92	1.41	2.02	0.025	0.040	0.139
Actinobacteria/Eggerthellaceae	*Eggerthella sinensis*	0.23	0.12	0.37	0.090	0.081	0.002
Bacteroidetes/Bacteroidaceae	*Bacteroides coprocola*	0.10	0.00	0.13	0.013	0.417	0.140
	*Bacteroides eggerthii*	0.12	0.10	0.00	0.413	0.003	0.125
	*Bacteroides stercoris*	0.63	0.35	0.84	0.093	0.233	0.023
	*Bacteroides thetaiotaomicron*	0.16	0.04	0.30	0.006	0.246	0.093
	*Bacteroides uniformis*	0.71	0.16	0.39	0.000	0.076	0.110
	*Bacteroides vulgatus*	1.01	0.32	0.43	0.003	0.016	0.260
Bacteroidetes/Tannerellaceae	*Parabacteroides merdae*	0.35	0.17	0.12	0.023	0.009	0.246
Euryarchaeota/Methanobacteriaceae	*Methanobrevibacter smithii*	0.54	0.30	0.11	0.155	0.034	0.015
Firmicutes/Acidaminococcaceae	*Phascolarctobacterium succinates*	0.23	0.15	0.07	0.225	0.033	0.132
Firmicutes/Clostridiaceae	*Clostridium cadaveris*	0.53	1.01	3.57	0.207	0.000	0.006
	*Clostridium perfringens*	0.39	0.28	0.00	0.313	0.005	0.015
	*Sarcina maxima*	0.12	0.13	0.54	0.394	0.016	0.022
Firmicutes/Enterococcaceae	*Enterococcus durans*	0.01	0.09	0.17	0.048	0.163	0.327
	*Enterococcus lactis*	0.16	0.76	0.08	0.029	0.231	0.027
Firmicutes/Eubacteriaceae	*Eubacterium cylindroides*	0.17	0.00	0.00	0.033	0.033	-
Firmicutes/Lachnospiraceae	*Blautia coccoides*	1.99	2.07	3.19	0.408	0.005	0.013
	*Blautia hydrogenotrophica*	0.01	0.00	0.24	0.163	0.009	0.009
	*Blautia obeum*	0.84	0.65	1.43	0.238	0.069	0.003
	*Blautia wexlerae*	1.92	1.54	3.22	0.300	0.042	0.013
	*Coprococcus eutactus*	0.14	0.16	0.51	0.435	0.022	0.001
	*Dorea formicigenerans*	0.17	0.28	0.13	0.066	0.260	0.005
	*Lachnospira pectinoachiza*	0.47	0.48	0.67	0.469	0.024	0.041
	*Pseudobutyrivibrio xylanivorans*	0.05	0.08	0.30	0.275	0.008	0.029
	*Roseburia faecis*	0.63	0.38	1.91	0.133	0.011	0.004
Firmicutes/Lactobacillaceae	*Lactobacillus delbrueckii*	0.02	0.14	0.00	0.182	0.012	0.139
	*Lactobacillus equi*	0.12	0.33	0.00	0.052	0.142	0.020
	*Lactobacillus fermentum*	0.15	0.39	0.00	0.012	0.005	0.000
	*Lactobacillus gasseri*	0.86	0.59	0.05	0.250	0.019	0.020
	*Lactobacillus intermedius*	0.09	0.26	0.05	0.013	0.322	0.042
	*Lactobacillus oris*	0.18	0.40	0.00	0.095	0.063	0.010
	*Lactobacillus salivarius*	2.12	7.61	0.05	0.047	0.142	0.021
	*Lactobacillus taiwanensis*	0.32	0.31	0.02	0.482	0.020	0.013
	*Lactobacillus ultunensis*	0.04	0.15	0.00	0.206	0.014	0.124
Firmicutes/Proteinivoraceae	*Anaerobranca zavarzinii*	0.78	0.18	0.22	0.036	0.049	0.323
Firmicutes/Ruminococcaceae	*Faecalibacterium prausnitzii*	2.53	2.19	3.75	0.275	0.008	0.005
	*Ruminococcus callidus*	0.30	0.08	0.08	0.014	0.009	0.491
	*Ruminococcus gnavus*	0.70	1.00	0.52	0.016	0.088	0.002
Firmicutes/Streptococcaceae	*Lactococcus garvieae*	0.54	0.13	0.00	0.114	0.042	0.139
	*Streptococcus anginosus*	0.05	0.26	0.38	0.035	0.190	0.380
	*Streptococcus bovis*	0.22	0.38	0.03	0.227	0.001	0.048
	*Streptococcus milleri*	0.02	0.13	0.33	0.058	0.177	0.276
	*Streptococcus mutans*	0.16	0.36	0.00	0.118	0.025	0.022
	*Streptococcus parasanguinis*	0.32	0.35	0.02	0.390	0.000	0.001
	*Streptococcus sobrinus*	0.15	0.67	0.00	0.039	0.163	0.022
	*Streptococcus vestibularis*	4.80	5.04	1.46	0.432	0.021	0.010
Proteobacteria/Enterobacteriaceae	*Escherichia albertii*	3.18	2.96	0.20	0.404	0.032	0.013
Proteobacteria/Yersiniaceae	*Serratia entomophila*	0.25	0.31	0.01	0.214	0.040	0.009
Verrucomicrobia/Akkermansiaceae	*Akkermansia muciniphila*	2.25	1.82	0.96	0.356	0.014	0.218

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
