# Peer review of "Nutritional Therapy Modulates Intestinal Microbiota and Reduces Serum Levels of Total and Free Indoxyl Sulfate and P-Cresyl Sulfate in Chronic Kidney Disease (Medika Study)"

_jcm, 2019, doi:10.3390/jcm8091424_

Round 1

Reviewer 1 Report

This is a truly excellent and clinically relevant study on nutritive care for patients with CKD. Both the design of the study and the presentation have been impressively structured. Accordingly, discussion and the graphical presentation of the work is excellent.
It is an impressively positive contribution to nutritive therapy in CKD and should provide a relevant basis for further studies. Therefore, the publication of the text is to be approved in every respect.

Minor remarks:
1. In Table 2  in line 3 VLPD 68.8±12.3 should refer to Age and not BMI 

2. The authors point out that patients receiving VLPD received one tablet every 5 kg of patient's ideal body weight, delivered during breakfast, lunch and dinner. In this regard, a supplement (possibly even a picture) regarding the tablet load would be desirable. And a supplementary comment, why their intake obviously did not cause any major problems.

Author Response

The authors thank the reviewers for the kind and exciting words used for the Medika study.

1. We have made corrections for the BMI made in Table 2 in line 3 VLPD.

2. The authors point out that VLPD patients received one tablet per 5 kg of ideal patient body weight, distributed for breakfast, lunch and dinner; the relative tablet load was 14.2 ± 2.5 cpr / patients corresponding a range of 8-18 cpr.
Taking these tablets obviously did not cause any clinical problems.

3. Free diet is the diet carried out by the patient before being enrolled in the study. We have added a figure to better show the study design.

4. The patients who participated in the study were not on dialysis but had a chronic kidney disease

5. authors collected blood samples at the end of each diet regimen average of two groups and thus determined the effect of VLPD and MD on IS

6. we inserted the value of GFR in the abstract. The GFR was 21.6±13.2 ml/min

Best regards, Biagio di iorio for all authors

Reviewer 2 Report

In this article, authors evaluated the effect of two diet regimen (Mediterranean and very low protein diet, MEDIKA study) crossover study in CKD patients and found the beneficial effects in reducing serum indoxyl sulfate and p-cresyl sulfate. I have following concerns.

Did CKD patients were undergoing dialysis? what is the eGFR? It should be included in the manuscript.

In the abstract, you can include eGFR value along with CKD 3B-4.

In table 2, BMI in VLPD is 68.8? Is there typo error in the table? Some numbers are missing in FD and VLPD group for the first three rows?

Did authors collected blood samples at the end of each diet regimen and take average from two groups? and thus determined the effect of VLPD and MD on IS?

What is free diet? Are patients allowed to eat diet of their choice? It should be made clear in the abstract and introduction.

Author Response

(The authors gave the same response as above.)
